# In Vivo Renal Lipid Quantification by Accelerated Magnetic Resonance Spectroscopic Imaging at 3T: Feasibility and Reliability Study

**DOI:** 10.3390/metabo12050386

**Published:** 2022-04-23

**Authors:** Ahmad A. Alhulail, Mahsa Servati, Nathan Ooms, Oguz Akin, Alp Dincer, M. Albert Thomas, Ulrike Dydak, Uzay E. Emir

**Affiliations:** 1Department of Radiology and Medical Imaging, Prince Sattam bin Abdulaziz University, Al-Kharj 16278, Saudi Arabia; 2School of Health Sciences, Purdue University, West Lafayette, IN 47907, USA; mservati@purdue.edu (M.S.); naooms@purdue.edu (N.O.); udydak@purdue.edu (U.D.); 3Department of Radiology and Imaging Sciences, Indiana University School of Medicine, Indianapolis, IN 46202, USA; 4Department of Radiology, Memorial Sloan Kettering Cancer Center, New York, NY 10065, USA; akino@mskcc.org; 5Department of Radiology, School of Medicine, Acibadem Mehmet Ali Aydinlar University, Istanbul 34684, Turkey; adincer@acibadem.com; 6Center for Neuroradiological Applications and Research, Acibadem Mehmet Ali Aydinlar University, Istanbul 34684, Turkey; 7Department of Radiology, University of California Los Angeles, Los Angeles, CA 90095, USA; athomas@mednet.ucla.edu; 8Weldon School of Biomedical Engineering, Purdue University, West Lafayette, IN 47907, USA

**Keywords:** kidney, renal, lipid, fast MRSI

## Abstract

A reliable and practical renal-lipid quantification and imaging method is needed. Here, the feasibility of an accelerated MRSI method to map renal fat fractions (FF) at 3T and its repeatability were investigated. A 2D density-weighted concentric-ring-trajectory MRSI was used for accelerating the acquisition of 48 × 48 voxels (each of 0.25 mL spatial resolution) without respiratory navigation implementations. The data were collected over 512 complex-FID timepoints with a 1250 Hz spectral bandwidth. The MRSI sequence was designed with a metabolite-cycling technique for lipid–water separation. The in vivo repeatability performance of the sequence was assessed by conducting a test–reposition–retest study within healthy subjects. The coefficient of variation (CV) in the estimated FF from the test–retest measurements showed a high degree of repeatability of MRSI-FF (CV = 4.3 ± 2.5%). Additionally, the matching level of the spectral signature within the same anatomical region was also investigated, and their intrasubject repeatability was also high, with a small standard deviation (8.1 ± 6.4%). The MRSI acquisition duration was ~3 min only. The proposed MRSI technique can be a reliable technique to quantify and map renal metabolites within a clinically acceptable scan time at 3T that supports the future application of this technique for the non-invasive characterization of heterogeneous renal diseases and tumors.

## 1. Introduction

The accumulation of lipids within and around kidney tissues has been linked to different renal pathophysiologies [1,2,3,4]. Recently, it was suggested that fatty kidney disease deserves designation as a specific clinical entity similar to fatty liver disease [5]. With this increase in interest in renal lipids, a non-invasive in vivo method to investigate their accumulation levels and locations is needed. Proton magnetic resonance imaging (MRI) and spectroscopy (MRS) methods have shown their potential to assess lipid-related kidney diseases, such as diabetic nephropathy, using fat fraction (FF) quantification [6,7,8,9].

Different MRI techniques for detecting abdominal lipid have been introduced [10,11,12,13,14]. However, these imaging methods provide summed fat fraction information, as they cannot differentiate lipid components (triglyceride fatty acids). Alternatively, the indirect detection of intracellular lipid (since it is considered a biomarker of renal cell carcinoma (RCC)) has been previously tried by an MRI method based on the signal drop in out-of-phase T1-weighted images [15,16,17]. This limitation of MRI can be addressed by implementing MRS, which can directly identify specific fatty acids and other metabolites, such as choline and lactate that can help in differentiating and grading RCC [18,19,20,21,22,23]. Due to the alteration in lipid content within the tumor cells of RCC, differentiation from other histological subtypes could be feasible by detecting intracellular lipid contents. For example, the elevation of the amount of lipid peak resonating at 1.3 ppm was used to discriminate clear cell RCC from non-clear cell RCC subtypes, which have less amount of this fatty acid [19]. This suggests that MRS could be useful for RCC characterization and tumor grading. However, renal MRS remains technically challenging. Although single-voxel MRS can differentiate lipid peaks in the kidney, it does not provide their signal distributions within large heterogeneous tumors. Conventional magnetic resonance spectroscopic imaging (MRSI) addresses this limitation by delivering spatially resolved spectra over many voxels but requiring a long acquisition time [24,25,26].

We recently demonstrated a high-resolution, density-weighted concentric ring trajectory (DW-CRT) metabolite cycling (MC) free induction decay (FID) MRSI acquisition technique to provide the spatially resolved musculoskeletal water and lipid spectra simultaneously [27]. In this work, our major goal was to investigate the feasibility of this accelerated MRSI acquisition to acquire reliable quantitative renal data in healthy volunteers with the intent to establish the spectral signature of the lipid composition of healthy renal tissues, which will be used as a future tool for the non-invasive characterization of renal diseases.

## 2. Results

The MRSI data that were collected in 3 min and 16 s were used to calculate FF maps. Examples of these MRSI FF maps are shown in Figure 1, where they were overlaid over their corresponding structural MRI images.

MRSI renal fat quantification per subject and the repeatability results were summarized in Table 1. The calculated mean CV was 4.3 ± 2.5%, representing excellent repeatability.

As shown in Figure 2, the comparison between the spectral signature from the repeated scans within the same anatomical region (kidney-cortex) showed a high consistency between the scans with high intrasubject repeatability of spectral signature (CV = 8.1 ± 6.4%).

The structural images produced by MRSI also provided general anatomical features comparable to MRI structural images, but with fewer details (Figure 3).

## 3. Discussion

In this work, our accelerated MRSI technique was evaluated for assessing renal fat contents at clinically available magnetic strength (3T). The MRSI method showed promising results. High reliability of fat-fraction quantification (CV < 5%) and good imaging abilities (anatomical representation) were demonstrated. The signature of lipid spectra from the same kidney region was also consistent between the scan sessions. MRSI data acquisition was completed within about 3 min, which is relatively short compared to the most common MRSI techniques. Based on our previous experiment [28], the DW-CRT trajectory not only improves SNR and reduces side lobes but also offers time efficiency compared with EPSI and conventional MRSI. DW-CRT achieves this by simultaneously sampling *k*-space in *k_x_* and *k_y_* directions [29]. In contrast, a conventional MRSI acquisition time is given by (*k_x_* points × *k_y_* points × TR), resulting in a longer scan time.

Although the MRSI was acquired without respiratory gating, the image and quantitative data were of good quality. Scanning without respiratory gating helped to maintain a short acquisition time. The main factor contributing to achieving good results without respiratory gating is related to how data were collected and post-processed. Each data set was composed of in-phase and out-of-phase spectra. The prominent water peaks within these spectra were matched to compensate for potential motions. Due to the high SNR water peak in the high-resolution MRSI voxel, non-water-suppressed metabolite-cycling MRSI can detect the frequency changes induced by motions. Thus, non-water-suppressed metabolite-cycling MRSI enables voxel-wise single-scan frequency, phase, and eddy current correction of metabolite spectra before subtraction, resulting in improved spectral quality [30]. However, this requires good shimming to reduce the peak’s width, improving the water-peak matching process. In this process, the average of the measured spectral linewidth was 24.5 ± 0.4 Hz, which was enough to achieve good outcomes. Although the technique provided a high degree of reproducibility of fat fraction, it would be interesting to acquire additional data sets with a respiratory gating method and to compare their results in the future. The breath-holding effects on the spectral quality of visible metabolites have been investigated in a previous MRS study [31]. In this previous study, the SNRs of the peaks of lipids and trimethylamine (TMA) moiety of choline metabolites were improved with breath-holding techniques, as less contamination from the surrounding tissue occurred. It is worth mentioning that this improvement has been observed in a large single voxel (8 mL), which suffers from more contamination if compared with the smaller voxels (0.25 mL) used in our method. In the same study, to employ the breath-holding approach with multivoxel spectroscopy, they suggested filling the k-space in segments to allow the patient to re-breathe. Although this segmentation of acquisition can allow a breath-holding approach, it prolongs the scan session’s duration. Thus, the implemented post-processing self-motion correction and smaller voxel sizes can promote our proposed method as a good alternative to improve data quality and to reduce scan times.

The MRSI images provided structural information that is sufficient to determine anatomical landmarks. For instance, as shown in Figure 3, one can identify kidneys and liver within the MRSI, which is in good agreement with its corresponding MRI image. However, the anatomical detail is not as good as what could be obtained with dedicated MRI sequences due to the lower MRSI spatial resolution, which is a standard limitation of most available MRSI sequences.

The exact MRSI sequence was previously tested on muscles and provided high-quality quantification results [27]. Here, we tested it on a more challenging area (moving and heterogeneous). In addition to extending the practicality of the sequence by granting more applications through the body regions, we decided to evaluate the technique on the kidney because of the clear need. According to the published reports in the field, there is some heterogeneity among studies regarding the mechanisms, consequences, and localization of renal lipid accumulation in the kidney, with a few in vivo studies performed on humans [1]. Additionally, the importance of metabolic imaging as a potential biomarker and research aid has been expressed in earlier publications [2]. Moreover, a need for a reliable MR spectroscopy method to quantify triglycerides in kidney structures was also expressed in other studies [32]. Although single-voxel MRS showed its powerful ability to provide unique information that can help diagnose many health disorders, it still faces several challenges. Some limitations of renal-MRS include its relatively low spatial resolution and the difficulty of assessing large heterogeneous tumors [18]. For instance, in addition to lipid fatty acids, MR spectroscopy methods allow gathering extra information about other metabolites such as choline, which was also used as a biomarker of RCC in the past [20]. However, the choline peak was clearer in relatively larger tumors, which returned to the potential volume effect factor that overwhelmed the choline peak [20]. Nevertheless, the signature of metabolites in renal tissues is different between the cortex and medulla, as shown in a previous ex vivo study [33]. This anatomical difference needs a higher spatial resolution than what is used in conventional MRS techniques. Therefore, employing MR spectroscopic imaging techniques that can provide the opportunity to evaluate large heterogeneous tumors requires a higher spatial resolution (≥what was used in this work). Accordingly, our proposed MRSI method can facilitate the non-invasive acquisition of human kidney data to provide a clearer idea about renal lipid’s role in pathophysiology. In addition to differentiating and grading RCC tumors, another potential application of the proposed renal-MRSI can include the diabetic kidney, which has been evaluated before using the MRS approach [9].

Although Dixon imaging methods can generate FF maps of a higher spatial resolution and usually within a short scan time while covering a larger anatomical FOV, MRS provides insight into the metabolism that is not achievable by other noninvasive methods [34]. Additionally, MRS is considered more accurate and used as the gold-stranded MR method to quantify FF, as it directly measures fat and water peaks [35]. Compared to Dixon, MRS/I methods can differentiate the signal of different fatty acids. In our study, we showed at least three peaks up-field the water peak (see Figure 2). In Dixon, these lipid peaks are summed up (undifferentiated). Several studies showed the importance of differentiating fatty acids peaks, as some individual peaks can be a biomarker of specific diseases. For example, the methylene lipid group (CH_2_)_n_ is linked to arterial stiffness [36], while the peak of the intramyocellular methylene (IMCL(CH_2_)_n_) is used as a biomarker for insulin resistivity [37,38] and mitochondrial disorder MELAS [39]. In kidneys, the ratio of free cholesterol and unsaturated fatty acid to saturated fatty acid at 1.3 ppm was suggested as a biomarker for metastatic RCC, which might be helpful in post-therapy monitoring [21]. The ratio between the renal lipid peak at 0.9 ppm over the lipid peak at 1.3 ppm was also suggested to differentiate between patients with RCC, renal infarct, renal tuberculosis, and healthy volunteers [23]. Moreover, the amount of lipid peak resonating at 1.3 ppm was used to discriminate the clear cell from non-clear cell RCC histologic subtypes [19]. The spectral signature was also suggested to differentiate the grade of RCC [22]. In a recent study, the renal triglyceride spectrum in type 2 Diabetes Mellitus patients was used to assess glycemic control influences [40]. This infers the possibility of using MR spectroscopy to evaluate glucose-lowering treatments.

Additionally, other metabolites such as choline and lactate can be detected by MRS/I techniques that are inaccessible to available imaging techniques. The choline peak detection in malignant renal tissues has been demonstrated in previous MRS studies and approved by histology [20]. The lactate peak was observed in patients with a tumor at an advanced stage, promoting it as a staging biomarker [22]. These peaks were not reported in this work, as the study was performed with healthy volunteers. However, if MRS has already detected choline and lactate signals, there are no reasons to expect that MRSI will not detect these metabolites. The only difference between the MRS and MRSI will be the ability to generate maps for each individual detected peak.

In terms of the accuracy of MRS methods, a recent study showed a high correlation between the quantified renal triglyceride content measured by 1H-MRS and the biopsy [9]. In a previous study, we performed a phantom study to evaluate the accuracy of our proposed MRSI and compared its results to a Dixon method, and a higher MRSI quantification accuracy was found [27].

The proposed MRSI method was able to detect the important lipid peaks that were used as biomarkers in previous MRS studies and are detectable in healthy subjects. Since the main goal of this study was to evaluate the feasibility of our proposed accelerated DW-CRT MRSI technique and its reliability for scanning kidneys, we preferred to conduct the study with healthy volunteers. In the future, we hope to use the proposed methods to assess the wide variety of renal abnormalities.

## 4. Materials and Methods

### 4.1. Human Subjects

In vivo abdominal MRIs were performed on five healthy volunteers (four males and one female; average age 31 ± 5 years; body mass index (BMI) = 25 ± 4 kg/m^2^). Informed consent was obtained from all subjects involved in the study before they participated in the study. The study was conducted following the guidelines of the institutional review board of Purdue University (protocol code 1102010525 on 24 January 2020).

### 4.2. Test–Retest Study

To evaluate the repeatability of the kidney-MRSI method, test–retest studies were conducted. The studies were performed on a 3 Tesla Siemens Prisma scanner (Siemens Healthineers, Erlangen, Germany). Subjects were asked to lie on a spine coil in a head-first supine position before a flexible coil (18-channel body) was placed above their abdominal region. The dedicated coils were used instead of the scanner integrated coil to improve the signal-to-noise ratio (SNR).

The acquisition protocol included two sequences: (1) a high-resolution T2-HASTE MRI sequence to provide structural reference images and (2) the proposed DW-CRT [41] MC FID-MRSI acquisition, which is used for fat fraction quantification [27].

The high-resolution T2-HASTE MRI reference images were acquired with TE/TR = 82/1200 ms, FA = 150°, number of averages = 1, spatial resolution = 0.9 × 0.9 × 4 mm^3^, FOV = 280 × 280 mm^2^, and echo train length = 83.

DW-CRT MRSI was prescribed using a Hanning window and the following parameters: acquisition delay = 4 ms, TR = 1000 ms, FA = 90°, number of averages = 1, FOV = 240 × 240 mm^2^ (one slice), slice thickness = 10 mm, matrix size = 48 × 48, extractable voxel resolution = 5 × 5 × 10 mm^3^ (0.25 mL nominal spatial resolution), TA = 192 s, number of rings = 24, points-per-ring = 64, temporal samples = 512, spatial interleaves = 4, time acquire = 96 s, and spectral bandwidth = 1250 Hz. No respiration navigation/triggering was used. This resulted in total acquisition duration of 3 min and 16 s. To enhance static field (B_0_) homogeneity, the left kidney area was shimmed before acquiring MRSI data. The typical full width at half maximum (FWHM) was 24.5 ± 0.4 Hz.

After a 30 min break outside the scanner, the subject returned to the scanner table and was repositioned before the repeat scan was acquired using the same scanning protocol. For repeatability purposes, MRSI data were obtained from an axial slice that demonstrated the same anatomy, marked by the kidney hilum (Figure 1).

### 4.3. MRSI Post-Processing

MRSI data were reconstructed and post-processed offline in MATLAB (MathWorks, Natick, MA, USA). Gridding and fast Fourier transform were performed using the nonuniform fast Fourier transform method [42] and without post hoc density compensation, as DW-CRT is already weighted by design [28]. B_0_ inhomogeneity was corrected by calculating the ΔB_0_ maps described in our previous work [43]. Here, the ΔB_0_ maps were calculated based on the first 2 MRSI phase-unwrapped images (TE1 = 4 ms and TE2 = 4.8 ms). The voxel-wise frequency and phase corrections were performed using cross-correlation and least-square fit algorithms, respectively, as described in Emir et al. [30]. The FIDs were smoothed using a Gaussian filter of 250 ms timing parameter and zero filling to 1024 time points. Next, the water-only and metabolite-only spectra were created by summing and subtracting the alternating FIDs, respectively, as described in Alhulail et al. [27].

### 4.4. Fat Fraction Quantification and Mapping

To estimate the signal under each spectral peak, spectral fitting was performed using LCModel [44]. An example of fitted spectra can be found in Figure 4. The integrated signals of each fitted lipid peak (between 0.8 and 1.7 ppm) and water peak were used to calculate the percentage of FF as follows.
(1)$$\mathrm{FF}=\frac{Lipid\;signal}{Total\;of\;lipid\;\&\;water\;signals}\times 100$$

To generate quantitative FF maps, the preceding process was performed for all voxels of the left kidney.

### 4.5. ROI Assignment and Statistical Analysis

The FF maps were first co-registered to their corresponding MRI images, which provide more precise structural details (Figure 1). Next, to assess the quantification repeatability, regions of interest (ROIs) were carefully drawn to cover several MRSI voxels only within the cortex region (to reduce anatomical variations) of the left kidney (Figure 2). Finally, the intra-subject coefficients of variation (CV) of the ROI’s FF were used to evaluate the repeatability of the MRSI outcomes.

## 5. Conclusions

The 2D density-weighted concentric ring trajectory MRSI is a reliable non-invasive method to quantify and map renal fat fractions. In addition, it provides a promising tool to further evaluate various renal diseases, such as diabetic kidney and renal tumors with their subtypes.

## Figures and Tables

**Figure 1 metabolites-12-00386-f001:**
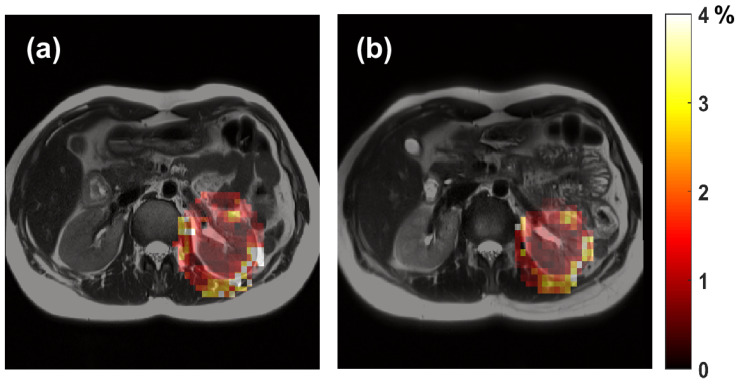
Representative data from: (**a**) the baseline; (**b**) repeated scans. The kidney hilum was used as an anatomical marker to acquire data from the same axial slice. The color-coded area is the coregistered MRSI fat-fraction map (masked about the left kidney) overlaid over its corresponding structural MRI image.

**Figure 2 metabolites-12-00386-f002:**
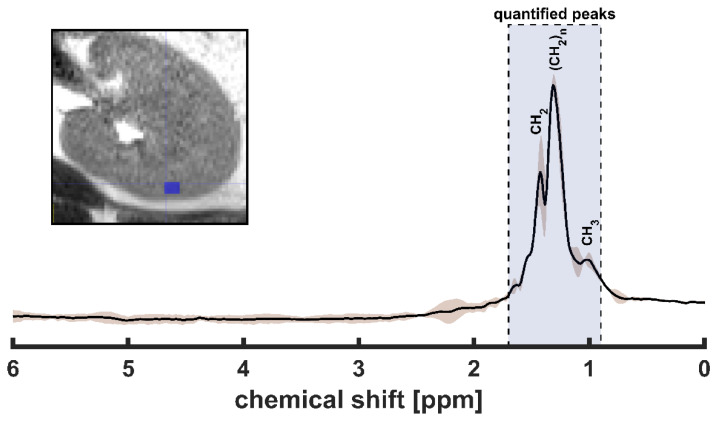
The spectral signature from the repeated scans within the kidney cortex. The image on the corner shows the region of interest (dark blue box) where the spectra were evaluated. The solid black line represents the mean signal, and the brown shade represents the standard deviation of the test–retest signal acquired from the same anatomical region. The blue shade highlights the peak’s bandwidth that has been covered to quantify the fat fraction. The labeled lipid peaks represent fatty acids of different saturation (CH_3_ at 0.9 ppm, (CH_2_)n at 1.3 ppm, and CH_2_ around 1.6 ppm).

**Figure 3 metabolites-12-00386-f003:**
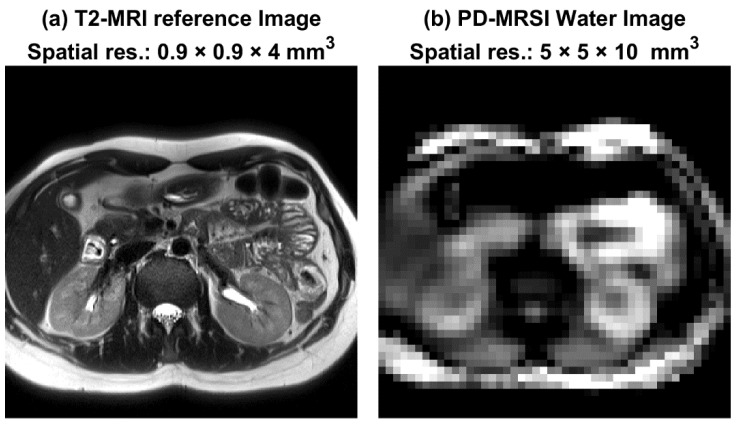
Example of anatomical image outputs. (**a**) A T2-MRI anatomical image (0.9 × 0.9 × 4 mm^3^); (**b**) The PD-water-only-MRSI anatomical image (5 × 5 × 10 mm^3^) for the same subject. Although spatial resolution and contrast are different (due to the variation in sequence parameters), similar structural details of the anatomy were demonstrated by both sequences.

**Figure 4 metabolites-12-00386-f004:**
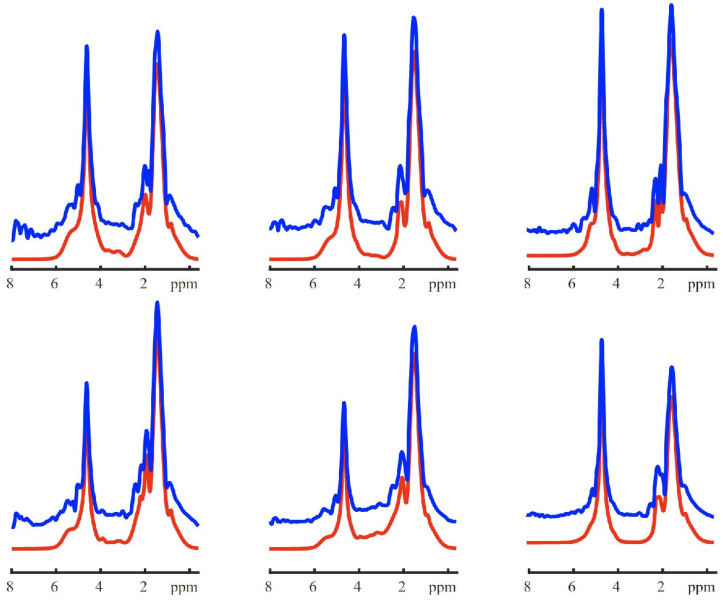
Example of lipid-only renal spectra fitting. Six spectra from adjacent voxels within the kidney are demonstrated. The blue lines represent the MRSI spectra, and the red lines represent their fit. The vertical offset between the blue and red line is due to the baseline correction implemented during fitting.

**Table 1 metabolites-12-00386-t001:** Fat quantification and its repeatability results.

Subject	Mean FF (%)	CV (%)
1	1.01 ± 0.05	4.90
2	1.60 ± 0.02	1.30
3	1.11 ± 0.06	5.80
4	1.69 ± 0.03	2.00
5	2.00 ± 0.15	7.40

FF, fat-fraction; CV, coefficient of variation.

## Data Availability

All data, tables, and figures in this manuscript are original, and data are available upon request from the corresponding authors, as it has not been uploaded to an online database.

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
