# Peer review of "In Vivo Renal Lipid Quantification by Accelerated Magnetic Resonance Spectroscopic Imaging at 3T: Feasibility and Reliability Study"

_metabolites, 2022, doi:10.3390/metabo12050386_

Round 1

Reviewer 1 Report

The manuscript is well written and presented. The study is well-conducted. The Authors should be commended for their work. I would recommend some minor checks regarding English grammar and structure.

Author Response

We thank the Reviewer for the efforts and time in reviewing our manuscript and for the overall positive comments.

To improve the English grammar and structure, we asked an English-speaking colleague to check the manuscript. Accordingly, we made some corrections to the revised manuscript.

Reviewer 2 Report

Dear authors,

Thanks for allowing us to review this manuscript by Alhulail et al., entitled "In Vivo Renal Lipid Quantification by Accelerated Magnetic Resonance Spectroscopic Imaging at 3T: Feasibility and Reliability Study". We appreciated this opportunity.

This work evaluated a novel application of DW-CRT MRSI techniques for rapid kidney 1H-MRSI. The data showed accurate quantification of renal lipids and good repeatability. While the given imaging sequence/technique itself has previously been reported by the same authors, applying this on characterizing renal fat content is novel and biologically important, and may improve future research of diabetic nephropathy, metabolic renal diseases as well as renal malignancies. Overall, we find the Methods and Results generally sound.

With that said, we felt the strengths of MRS over standard fat-water imaging need to be further clarified in the context of renal lipid characterization -- as this study sought to address a "clear [unmet] need" (Line 150). Our specific comments are as follows,

Major
1. The biomarker of choice in the manuscript was fat fractions (FF). However, conventional fat-water imaging such as mDixon can readily create fat-only, water only, and FF maps with higher spatial resolution (and oftentimes coverage) than MRSI, and doing so within a single breathhold. In this sense, we were wondering whether it can justified that MRSI provides unique information inaccessible to fat-water imaging?

Based on the acquired data, can different species of lipids (e.g. triglycerides vs others), or metabolites such as lactate, choline (as alluded to in Line 55) be individually detected and quantified via the LCModel fitting (Line 223)? And if so -- how, specifically, would such capability benefit characterization of renal diseases? (For instance, choline/NAA informs on glioblastoma activity - is there an analogy for metabolic kidney diseases or tumors?)
One idea is to generate spatial maps of either individual metabolites, or any calculated biomarkers of specific interest to the diseases mentioned in the Introduction section. Showing the feasibility to quantify these biomarkers, beyond FF, could be a way to justify the value of fast renal MRSI techniques over conventional imaging.

Alternatively, was the definition of FF in this manuscript different from the conventional one? In the latter case, the differences should be clarified, and the utility of the alternate definition for renal disease characterization should be briefly reviewed.

2. We felt that it may also beneficial to discuss the choice of excluding fat-water imaging from the experimental protocol, and perhaps whether a head-to-head comparison of the FF maps between mDixon and the proposed MRSI would further validate the quantitative accuracy of lipid levels in the kidneys. Would MRSI be more accurate? We appreciate the fact that it may be challenging to make such comparisons retrospectively, but a brief discussion may help the readers better understand the context.

Minor
3. Line 57: Here "other histological subtypes" was mentioned. Which subtypes are being referred to? How are RCC subtypes such as ccRCC, chromophobe or papillary RCC differentiated via the amount or form of lipids? It would be good to briefly give the specifics, and highlight whether the results of this study showed the technical feasibility of detecting/quantifying these relevant metabolites.

4. Line 74 & 114: The total scan time was ~3 minutes for kidney imaging using the proposed method. What was the acceleration factor against conventional MRSI and how long would it have taken if conventionally-encoded MRSI were used?

5. Line 120-122: "Each data set ... compensate for the potential motions." How matching spectra was done, and why it reduces motion artifact could be obscure to unfamiliar readers. We'd suggest either describe this briefly in the Methods, or provide references if it has been described before.

6. Line 164: "... signature of metabolites in normal and RCC tissues is different between the cortex and medulla." This sentence appears to be unclear. Is the metabolic signature different between disease vs normal, cortex vs medulla, or both?

7. Line 165-167: "This anatomical difference ... higher spatial resolution is required." This is also somewhat vague. Is this intended to acknowledge a limitation of the proposed technique, or to highlight its strength?

8. Line 186: An abdominal array coil was used. Is the proposed technique compatible with parallel imaging? Would this be able to further accelerate the acquisition?

9. Line 195: Here the FOV was provided. How many total slices were acquired, and what was the slice thickness?

10. Line 199 & 212: What was the typical intra-voxel B0 heterogeneity in the kidney? Was the observed linewidth consistent with ΔB0?

11. Figure 1: What was the unit of the y-axis on the colorbar?

12. Figure 4: From the image, the red and blue curves seemed to differ by an offset. Was baseline correction used in the fitting?

Best regards,

Round 2

Reviewer 2 Report

No additional comments.